# Fixup Initialization: Residual Learning Without Normalization

**Hongyi Zhang**[*]
MIT
hongyiz@mit.edu

**Yann N. Dauphin**[†]
Google Brain
yann@dauphin.io

**Tengyu Ma**[‡]
Stanford University
tengyuma@stanford.edu

## Abstract

Normalization layers are a staple in state-of-the-art deep neural network architectures. They are widely believed to stabilize training, enable higher learning rate, accelerate convergence and improve generalization, though the reason for their effectiveness is still an active research topic. In this work, we challenge the commonly-held beliefs by showing that *none* of the perceived benefits is unique to normalization. Specifically, we propose *fixed-update initialization* (Fixup), an initialization motivated by solving the exploding and vanishing gradient problem at the beginning of training via properly rescaling a standard initialization. We find training residual networks with Fixup to be as stable as training with normalization — even for networks with 10,000 layers. Furthermore, with proper regularization, Fixup enables residual networks *without* normalization to achieve state-of-the-art performance in image classification and machine translation.

## 1 Introduction

Artificial intelligence applications have witnessed major advances in recent years. At the core of this revolution is the development of novel neural network models and their training techniques. For example, since the landmark work of He et al. (2016), most of the state-of-the-art image recognition systems are built upon a deep stack of network blocks consisting of convolutional layers and additive skip connections, with some normalization mechanism (e.g., batch normalization (Ioffe & Szegedy, 2015)) to facilitate training and generalization. Besides image classification, various normalization techniques (Ulyanov et al., 2016; Ba et al., 2016; Salimans & Kingma, 2016; Wu & He, 2018) have been found essential to achieving good performance on other tasks, such as machine translation (Vaswani et al., 2017) and generative modeling (Zhu et al., 2017). They are widely believed to have multiple benefits for training very deep neural networks, including stabilizing learning, enabling higher learning rate, accelerating convergence, and improving generalization.

Despite the enormous empirical success of training deep networks with normalization, and recent progress on understanding the working of batch normalization (Santurkar et al., 2018), there is currently no general consensus on why these normalization techniques help training residual neural networks. Intrigued by this topic, in this work we study

(i) *without* normalization, can a deep residual network be trained reliably? (And if so,)
(ii) *without* normalization, can a deep residual network be trained with the same learning rate, converge at the same speed, and generalize equally well (or even better)?

Perhaps surprisingly, we find the answers to both questions are *Yes*. In particular, we show:

- **Why normalization helps training.** We derive a lower bound for the gradient norm of a residual network at initialization, which explains why with standard initializations, normalization techniques are *essential* for training deep residual networks at maximal learning rate. (Section 2)

---

[*]Work done at Facebook. Equal contribution.
[†]Work done at Facebook. Equal contribution.
[‡]Work done at Facebook.

- **Training without normalization.** We propose Fixup, a method that rescales the standard initialization of residual branches by adjusting for the network architecture. Fixup enables training very deep residual networks stably at maximal learning rate without normalization. (Section 3)

- **Image classification.** We apply Fixup to replace batch normalization on image classification benchmarks CIFAR-10 (with Wide-ResNet) and ImageNet (with ResNet), and find Fixup with proper regularization matches the well-tuned baseline trained with normalization. (Section 4.2)

- **Machine translation.** We apply Fixup to replace layer normalization on machine translation benchmarks IWSLT and WMT using the Transformer model, and find it outperforms the baseline and achieves new state-of-the-art results on the same architecture. (Section 4.3)

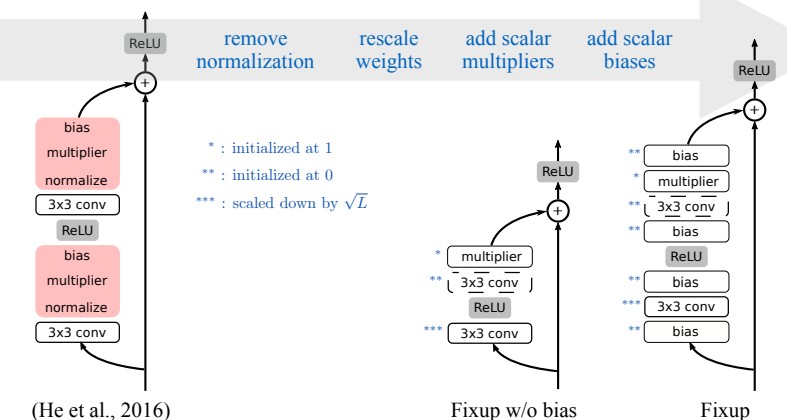

Figure 1: **Left:** ResNet basic block. Batch normalization (Ioffe & Szegedy, 2015) layers are marked in red. **Middle:** A simple network block that trains stably when stacked together. **Right:** Fixup further improves by adding bias parameters. (See Section 3 for details.)

In the remaining of this paper, we first analyze the exploding gradient problem of residual networks at initialization in Section 2. To solve this problem, we develop Fixup in Section 3. In Section 4 we quantify the properties of Fixup and compare it against state-of-the-art normalization methods on real world benchmarks. A comparison with related work is presented in Section 5.

## 2 PROBLEM: RESNET WITH STANDARD INITIALIZATIONS LEAD TO EXPLODING GRADIENTS

Standard initialization methods (Glorot & Bengio, 2010; He et al., 2015; Xiao et al., 2018) attempt to set the initial parameters of the network such that the activations neither vanish nor explode. Unfortunately, it has been observed that without normalization techniques such as BatchNorm they do not account properly for the effect of residual connections and this causes exploding gradients. Balduzzi et al. (2017) characterizes this problem for ReLU networks, and we will generalize this to residual networks with positively homogenous activation functions. A plain (i.e. without normalization layers) ResNet with residual blocks $\{F_1, \ldots, F_L\}$ and input $\mathbf{x}_0$ computes the activations as

$$\mathbf{x}_l = \mathbf{x}_0 + \sum_{i=0}^{l-1} F_i(\mathbf{x}_i). \tag{1}$$

**ResNet output variance grows exponentially with depth.** Here we only consider the initialization, view the input $\mathbf{x}_0$ as fixed, and consider the randomness of the weight initialization. We analyze the variance of each layer $\mathbf{x}_l$, denoted by $\mathrm{Var}[\mathbf{x}_l]$ (which is technically defined as the sum of the variance of all the coordinates of $\mathbf{x}_l$.) For simplicity we assume the blocks are initialized to be zero mean, i.e., $\mathbb{E}[F_l(\mathbf{x}_l) \mid \mathbf{x}_l] = 0$. By $\mathbf{x}_{l+1} = \mathbf{x}_l + F_l(\mathbf{x}_l)$, and the law of total variance, we have $\mathrm{Var}[\mathbf{x}_{l+1}] = \mathbb{E}[\mathrm{Var}[F(\mathbf{x}_l)|\mathbf{x}_l]] + \mathrm{Var}(\mathbf{x}_l)$. Resnet structure prevents $\mathbf{x}_l$ from vanishing by forcing the variance to grow with depth, i.e. $\mathrm{Var}[\mathbf{x}_l] < \mathrm{Var}[\mathbf{x}_{l+1}]$ if $\mathbb{E}[\mathrm{Var}[F(\mathbf{x}_l)|\mathbf{x}_l]] > 0$. Yet, combined with initialization methods such as He et al. (2015), the output variance of each residual branch

$\text{Var}[F_l(\mathbf{x}_l)|\mathbf{x}_l]$ will be about the same as its input variance $\text{Var}[\mathbf{x}_l]$, and thus $\text{Var}[\mathbf{x}_{l+1}] \approx 2\text{Var}[\mathbf{x}_l]$. This causes the output variance to explode exponentially with depth without normalization (Hanin & Rolnick, 2018) for positively homogeneous blocks (see Definition 1). This is detrimental to learning because it can in turn cause gradient explosion.

As we will show, at initialization, the gradient norm of certain activations and weight tensors is *lower bounded* by the cross-entropy loss up to some constant. Intuitively, this implies that blowup in the logits will cause gradient explosion. Our result applies to convolutional and linear weights in a neural network with ReLU nonlinearity (e.g., feed-forward network, CNN), possibly with skip connections (e.g., ResNet, DenseNet), but without any normalization.

Our analysis utilizes properties of positively homogeneous functions, which we now introduce.

**Definition 1** (positively homogeneous function of first degree). A function $f : \mathbb{R}^m \to \mathbb{R}^n$ is called *positively homogeneous (of first degree)* (p.h.) if for any input $\mathbf{x} \in \mathbb{R}^m$ and $\alpha > 0$, $f(\alpha\mathbf{x}) = \alpha f(\mathbf{x})$.

**Definition 2** (positively homogeneous set of first degree). Let $\theta = \{\theta_i\}_{i \in S}$ be the set of parameters of $f(\mathbf{x})$ and $\theta_{ph} = \{\theta_i\}_{i \in S_{ph} \subset S}$. We call $\theta_{ph}$ a *positively homogeneous set (of first degree)* (p.h. set) if for any $\alpha > 0$, $f(\mathbf{x}; \theta \setminus \theta_{ph}, \alpha\theta_{ph}) = \alpha f(\mathbf{x}; \theta \setminus \theta_{ph}, \theta_{ph})$, where $\alpha\theta_{ph}$ denotes $\{\alpha\theta_i\}_{i \in S_{ph}}$.

Intuitively, a p.h. set is a set of parameters $\theta_{ph}$ in function $f$ such that for any fixed input $\mathbf{x}$ and fixed parameters $\theta \setminus \theta_{ph}$, $\bar{f}(\theta_{ph}) \triangleq f(\mathbf{x}; \theta \setminus \theta_{ph}, \theta_{ph})$ is a p.h. function.

Examples of p.h. functions are ubiquitous in neural networks, including various kinds of linear operations without bias (fully-connected (FC) and convolution layers, pooling, addition, concatenation and dropout etc.) as well as ReLU nonlinearity. Moreover, we have the following claim:

**Proposition 1.** *A function that is the composition of p.h. functions is itself p.h.*

We study classification problems with $c$ classes and the cross-entropy loss. We use $f$ to denote a neural network function except for the softmax layer. Cross-entropy loss is defined as $\ell(\mathbf{z}, \mathbf{y}) \triangleq -\mathbf{y}^T(\mathbf{z} - \texttt{logsumexp}(\mathbf{z}))$ where $\mathbf{y}$ is the one-hot label vector, $\mathbf{z} \triangleq f(\mathbf{x}) \in \mathbb{R}^c$ is the logits where $z_i$ denotes its $i$-th element, and $\texttt{logsumexp}(\mathbf{z}) \triangleq \log\left(\sum_{i \in [c]} \exp(z_i)\right)$. Consider a minibatch of training examples $\mathcal{D}_M = \{(\mathbf{x}^{(m)}, \mathbf{y}^{(m)})\}_{m=1}^M$ and the average cross-entropy loss $\ell_{\text{avg}}(\mathcal{D}_M) \triangleq \frac{1}{M}\sum_{m=1}^M \ell(f(\mathbf{x}^{(m)}), \mathbf{y}^{(m)})$, where we use $^{(m)}$ to index quantities referring to the $m$-th example. $\|\cdot\|$ denotes any valid norm. We only make the following assumptions about the network $f$:

1. $f$ is a sequential composition of network blocks $\{f_i\}_{i=1}^L$, i.e. $f(\mathbf{x}_0) = f_L(f_{L-1}(\ldots f_1(\mathbf{x}_0)))$, each of which is composed of p.h. functions.
2. Weight elements in the FC layer are i.i.d. sampled from a zero-mean symmetric distribution.

These assumptions hold at initialization if we remove all the normalization layers in a residual network with ReLU nonlinearity, assuming all the biases are initialized at 0.

Our results are summarized in the following two theorems, whose proofs are listed in the appendix:

**Theorem 1.** *Denote the input to the $i$-th block by $\mathbf{x}_{i-1}$. With Assumption 1, we have*

$$\left\|\frac{\partial\ell}{\partial\mathbf{x}_{i-1}}\right\| \geq \frac{\ell(\mathbf{z}, \mathbf{y}) - H(\mathbf{p})}{\|\mathbf{x}_{i-1}\|}, \tag{2}$$

*where $\mathbf{p}$ is the softmax probabilities and $H$ denotes the Shannon entropy.*

Since $H(\mathbf{p})$ is upper bounded by $\log(c)$ and $\|\mathbf{x}_{i-1}\|$ is small in the lower blocks, blowup in the loss will cause large gradient norm with respect to the lower block input. Our second theorem proves a lower bound on the gradient norm of a p.h. set in a network.

**Theorem 2.** *With Assumption 1, we have*

$$\left\|\frac{\partial\ell_{\text{avg}}}{\partial\theta_{ph}}\right\| \geq \frac{1}{M\|\theta_{ph}\|}\sum_{m=1}^M \ell(\mathbf{z}^{(m)}, \mathbf{y}^{(m)}) - H(\mathbf{p}^{(m)}) \triangleq G(\theta_{ph}). \tag{3}$$

*Furthermore, with Assumptions 1 and 2, we have*

$$\mathbb{E}G(\theta_{ph}) \geq \frac{\mathbb{E}[\max_{i \in [c]} z_i] - \log(c)}{\|\theta_{ph}\|}. \tag{4}$$

It remains to identify such p.h. sets in a neural network. In Figure 2 we provide three examples of p.h. sets in a ResNet without normalization. Theorem 2 suggests that these layers would suffer from the exploding gradient problem, if the logits $\mathbf{z}$ blow up at initialization, which unfortunately would occur in a ResNet without normalization if initialized in a traditional way. This motivates us to introduce a new initialization in the next section.

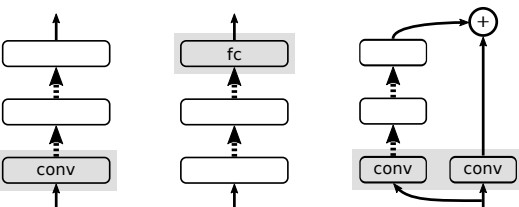

Figure 2: Examples of p.h. sets in a ResNet without normalization: (1) the first convolution layer before max pooling; (2) the fully connected layer before softmax; (3) the union of a spatial down-sampling layer in the backbone and a convolution layer in its corresponding residual branch.

## 3 FIXUP: UPDATE A RESIDUAL NETWORK $\Theta(\eta)$ PER SGD STEP

Our analysis in the previous section points out the failure mode of standard initializations for training deep residual network: the gradient norm of certain layers is in expectation lower bounded by a quantity that increases indefinitely with the network depth. However, escaping this failure mode does not necessarily lead us to successful training — after all, it is *the whole network as a function* that we care about, rather than a layer or a network block. In this section, we propose a top-down design of a new initialization that ensures proper update scale to the network function, by simply rescaling a standard initialization. To start, we denote the learning rate by $\eta$ and set our goal:

> $f(\mathbf{x}; \theta)$ *is updated by* $\Theta(\eta)$ *per SGD step after initialization as* $\eta \to 0$.
> That is, $\|\Delta f(\mathbf{x})\| = \Theta(\eta)$ where $\Delta f(\mathbf{x}) \triangleq f(\mathbf{x}; \theta - \eta \frac{\partial}{\partial \theta} \ell(f(\mathbf{x}), \mathbf{y})) - f(\mathbf{x}; \theta)$.

Put another way, our goal is to design an initialization such that SGD updates to the network function are in the right scale and independent of the depth.

We define the *Shortcut* as the shortest path from input to output in a residual network. The Shortcut is typically a shallow network with a few trainable layers.[1] We assume the Shortcut is initialized using a standard method, and focus on the initialization of the residual branches.

**Residual branches update the network in sync.** To start, we first make an important observation that the SGD update to each residual branch changes the network output in highly correlated directions. This implies that if a residual network has $L$ residual branches, then an SGD step to each residual branch should change the network output by $\Theta(\eta/L)$ on average to achieve an overall $\Theta(\eta)$ update. We defer the formal statement and its proof until Appendix B.1.

**Study of a scalar branch.** Next we study how to initialize a residual branch with $m$ layers so that its SGD update changes the network output by $\Theta(\eta/L)$. We assume $m$ is a small positive integer (e.g., 2 or 3). As we are only concerned about the scale of the update, it is sufficiently instructive to study the scalar case, i.e., $F(x) = (\prod_{i=1}^{m} a_i) x$ where $a_1, \ldots, a_m, x \in \mathbb{R}^+$. For example, the standard initialization methods typically initialize each layer so that the output (after nonlinear activation) preserves the input variance, which can be modeled as setting $\forall i \in [m], a_i = 1$. In turn, setting $a_i$ to a positive number other than 1 corresponds to rescaling the i-th layer by $a_i$.

Through deriving the constraints for $F(x)$ to make $\Theta(\eta/L)$ updates, we will also discover how to rescale the weight layers of a standard initialization as desired. In particular, we show the SGD

---

[1]For example, in the ResNet architecture (e.g., ResNet-50, ResNet-101 or ResNet-152) for ImageNet classification, the Shortcut is always a 6-layer network with five convolution layers and one fully-connected layer, irrespective of the total depth of the whole network.

update to $F(x)$ is $\Theta(\eta/L)$ *if and only if* the initialization satisfies the following constraint:

$$\left(\prod_{i \in [m] \setminus \{j\}} a_i\right) x = \Theta\left(\frac{1}{\sqrt{L}}\right), \quad \text{where} \quad j \in \arg\min_k a_k \tag{5}$$

We defer the derivation until Appendix B.2.

Equation (5) suggests new methods to initialize a residual branch through *rescaling the standard initialization of i-th layer in a residual branch by its corresponding scalar $a_i$*. For example, we could set $\forall i \in [m], a_i = L^{-\frac{1}{2m-2}}$. Alternatively, we could start the residual branch as a zero function by setting $a_m = 0$ and $\forall i \in [m-1], a_i = L^{-\frac{1}{2m-2}}$. In the second option, the residual branch does not need to "unlearn" its potentially bad random initial state, which can be beneficial for learning. Therefore, we use the latter option in our experiments, unless otherwise specified.

**The effects of biases and multipliers.** With proper rescaling of the weights in all the residual branches, a residual network is supposed to be updated by $\Theta(\eta)$ per SGD step — our goal is achieved. However, in order to match the training performance of a corresponding network with normalization, there are two more things to consider: biases and multipliers.

Using biases in the linear and convolution layers is a common practice. In normalization methods, bias and scale parameters are typically used to restore the representation power after normalization.[2] Intuitively, because the preferred input/output mean of a weight layer may be different from the preferred output/input mean of an activation layer, it also helps to insert bias terms in a residual network without normalization. Empirically, we find that inserting just one scalar bias before each weight layer and nonlinear activation layer significantly improves the training performance.

Multipliers scale the output of a residual branch, similar to the scale parameters in batch normalization. They have an interesting effect on the learning dynamics of weight layers in the same branch. Specifically, as the stochastic gradient of a layer is typically almost orthogonal to its weight, learning rate decay tends to cause the weight norm equilibrium to shrink when combined with L2 weight decay (van Laarhoven, 2017). In a branch with multipliers, this in turn causes the growth of the multipliers, increasing the effective learning rate of other layers. In particular, we observe that inserting just one scalar multiplier per residual branch mimics the weight norm dynamics of a network with normalization, and spares us the search of a new learning rate schedule.

Put together, we propose the following method to train residual networks without normalization:

> **Fixup initialization (or: How to train a deep residual network without normalization)**
>
> 1. Initialize the classification layer and the last layer of each residual branch to $0$.
> 2. Initialize every other layer using a standard method (e.g., He et al. (2015)), and scale only the weight layers inside residual branches by $L^{-\frac{1}{2m-2}}$.
> 3. Add a scalar multiplier (initialized at 1) in every branch and a scalar bias (initialized at 0) before each convolution, linear, and element-wise activation layer.

It is important to note that Rule 2 of Fixup is the essential part as predicted by Equation (5). Indeed, we observe that using Rule 2 alone is sufficient and necessary for training extremely deep residual networks. On the other hand, Rule 1 and Rule 3 make further improvements for training so as to match the performance of a residual network with normalization layers, as we explain in the above text.[3] We find ablation experiments confirm our claims (see Appendix C.1).

---

[2] For example, in batch normalization gamma and beta parameters are used to affine-transform the normalized activations per each channel.

[3] It is worth noting that the design of Fixup is a simplification of the common practice, in that we only introduce $O(K)$ parameters beyond convolution and linear weights (since we remove bias terms from convolution and linear layers), whereas the common practice includes $O(KC)$ (Ioffe & Szegedy, 2015; Salimans & Kingma, 2016) or $O(KCWH)$ (Ba et al., 2016) additional parameters, where $K$ is the number of layers, $C$ is the max number of channels per layer and $W, H$ are the spatial dimension of the largest feature maps.

Our initialization and network design is consistent with recent theoretical work Hardt & Ma (2016); Li et al. (2018), which, in much more simplified settings such as linearized residual nets and quadratic neural nets, propose that small initialization tend to stabilize optimization and help generalizaiton. However, our approach suggests that more delicate control of the scale of the initialization is beneficial.[4]

## 4 EXPERIMENTS

### 4.1 TRAINING AT INCREASING DEPTH

One of the key advatanges of BatchNorm is that it leads to fast training even for very deep models (Ioffe & Szegedy, 2015). Here we will determine if we can match this desirable property by relying only on proper initialization. We propose to evaluate how each method affects training very deep nets by *measuring the test accuracy after the first epoch as we increase depth*. In particular, we use the wide residual network (WRN) architecture with width 1 and the default weight decay $5e-4$ (Zagoruyko & Komodakis, 2016). We specifically use the default learning rate of $0.1$ because the ability to use high learning rates is considered to be important to the success of BatchNorm. We compare Fixup against three baseline methods — (1) rescale the output of each residual block by $\frac{1}{\sqrt{2}}$ (Balduzzi et al., 2017), (2) post-process an orthogonal initialization such that the output variance of each residual block is close to 1 (Layer-sequential unit-variance orthogonal initialization, or LSUV) (Mishkin & Matas, 2015), (3) batch normalization (Ioffe & Szegedy, 2015). We use the default batch size of 128 up to 1000 layers, with a batch size of 64 for 10,000 layers. We limit our budget of epochs to 1 due to the computational strain of evaluating models with up to 10,000 layers.

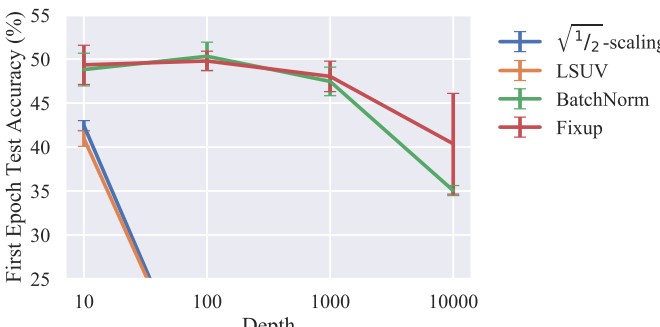

Figure 3: Depth of residual networks versus test accuracy at the first epoch for various methods on CIFAR-10 with the default BatchNorm learning rate. We observe that Fixup is able to train very deep networks with the same learning rate as batch normalization. (Higher is better.)

Figure 3 shows the test accuracy at the first epoch as depth increases. Observe that Fixup matches the performance of BatchNorm at the first epoch, even with 10,000 layers. LSUV and $\sqrt{1/2}$-scaling are not able to train with the same learning rate as BatchNorm past 100 layers.

### 4.2 IMAGE CLASSIFICATION

In this section, we evaluate the ability of Fixup to replace batch normalization in image classification applications. On the CIFAR-10 dataset, we first test on ResNet-110 (He et al., 2016) with default hyper-parameters; results are shown in Table 1. Fixup obtains 7% relative improvement in test error compared with standard initialization; however, we note a substantial difference in the difficulty of training. While network with Fixup is trained with the same learning rate and converge as fast as network with batch normalization, we fail to train a Xavier initialized ResNet-110 with 0.1x maximal learning rate.[5] The test error gap in Table 1 is likely due to the regularization effect of BatchNorm

---

[4]For example, learning rate smaller than our choice would also stabilize the training, but lead to lower convergence rate.

[5]Personal communication with the authors of (Shang et al., 2017) confirms our observation, and reveals that the Xavier initialized network need more epochs to converge.

rather than difficulty in optimization; when we train Fixup networks with better regularization, the test error gap disappears and we obtain state-of-the-art results on CIFAR-10 and SVHN without normalization layers (see Appendix C.2).

| Dataset | ResNet-110 | Normalization | Large $\eta$ | Test Error (%) |
|---------|------------|:-------------:|:------------:|---------------:|
| CIFAR-10 | w/ BatchNorm (He et al., 2016) | ✓ | ✓ | 6.61 |
| | w/ Xavier Init (Shang et al., 2017) | ✗ | ✗ | 7.78 |
| | w/ Fixup-init | ✗ | ✓ | 7.24 |

Table 1: Results on CIFAR-10 with ResNet-110 (mean/median of 5 runs; lower is better).

On the ImageNet dataset, we benchmark Fixup with the ResNet-50 and ResNet-101 architectures (He et al., 2016), trained for 100 epochs and 200 epochs respectively. Similar to our finding on the CIFAR-10 dataset, we observe that (1) training with Fixup is fast and stable with the default hyperparameters, (2) Fixup alone significantly improves the test error of standard initialization, and (3) there is a large test error gap between Fixup and BatchNorm. Further inspection reveals that Fixup initialized models obtain significantly lower training error compared with BatchNorm models (see Appendix C.3), i.e., Fixup suffers from overfitting. We therefore apply stronger regularization to the Fixup models using Mixup (Zhang et al., 2017). We find it is beneficial to reduce the learning rate of the scalar multiplier and bias by 10x when additional large regularization is used. Best Mixup coefficients are found through cross-validation: they are 0.2, 0.1 and 0.7 for BatchNorm, GroupNorm (Wu & He, 2018) and Fixup respectively. We present the results in Table 2, noting that with better regularization, the performance of Fixup is on par with GroupNorm.

| Model | Method | Normalization | Test Error (%) |
|-------|--------|:-------------:|---------------:|
| ResNet-50 | BatchNorm (Goyal et al., 2017) | | 23.6 |
| | BatchNorm + Mixup (Zhang et al., 2017) | ✓ | **23.3** |
| | GroupNorm + Mixup | | 23.9 |
| | Xavier Init (Shang et al., 2017) | | 31.5 |
| | Fixup-init | ✗ | 27.6 |
| | Fixup-init + Mixup | | **24.0** |
| ResNet-101 | BatchNorm (Zhang et al., 2017) | | 22.0 |
| | BatchNorm + Mixup (Zhang et al., 2017) | ✓ | **20.8** |
| | GroupNorm + Mixup | | 21.4 |
| | Fixup-init + Mixup | ✗ | 21.4 |

Table 2: ImageNet test results using the ResNet architecture. (Lower is better.)

### 4.3 MACHINE TRANSLATION

To demonstrate the generality of Fixup, we also apply it to replace layer normalization (Ba et al., 2016) in Transformer (Vaswani et al., 2017), a state-of-the-art neural network for machine translation. Specifically, we use the fairseq library (Gehring et al., 2017) and follow the Fixup template in Section 3 to modify the baseline model. We evaluate on two standard machine translation datasets, IWSLT German-English (de-en) and WMT English-German (en-de) following the setup of Ott et al. (2018). For the IWSLT de-en dataset, we cross-validate the dropout probability from $\{0.3, 0.4, 0.5, 0.6\}$ and find $0.5$ to be optimal for both Fixup and the LayerNorm baseline. For the WMT'16 en-de dataset, we use dropout probability $0.4$. All models are trained for 200k updates.

It was reported (Chen et al., 2018) that "Layer normalization is most critical to stabilize the training process... removing layer normalization results in unstable training runs". However we find training with Fixup to be very stable and as fast as the baseline model. Results are shown in Table 3. Surprisingly, we find the models do not suffer from overfitting when LayerNorm is replaced by Fixup, thanks to the strong regularization effect of dropout. Instead, Fixup matches or supersedes the state-of-the-art results using Transformer model on both datasets.

| Dataset | Model | Normalization | BLEU |
|---------|-------|:-------------:|------|
| IWSLT DE-EN | (Deng et al., 2018)
LayerNorm | ✓ | 33.1
34.2 |
| | Fixup-init | ✗ | **34.5** |
| WMT EN-DE | (Vaswani et al., 2017)
LayerNorm (Ott et al., 2018) | ✓ | 28.4
**29.3** |
| | Fixup-init | ✗ | **29.3** |

Table 3: Comparing Fixup vs. LayerNorm for machine translation tasks. (Higher is better.)

## 5 RELATED WORK

**Normalization methods.** Normalization methods have enabled training very deep residual networks, and are currently an essential building block of the most successful deep learning architectures. All normalization methods for training neural networks explicitly normalize (i.e. standardize) some component (activations or weights) through dividing activations or weights by some real number computed from its statistics and/or subtracting some real number activation statistics (typically the mean) from the activations.[6] In contrast, Fixup does not compute statistics (mean, variance or norm) at initialization or during any phase of training, hence is not a normalization method.

**Theoretical analysis of deep networks.** Training very deep neural networks is an important theoretical problem. Early works study the propagation of variance in the forward and backward pass for different activation functions (Glorot & Bengio, 2010; He et al., 2015).

Recently, the study of *dynamical isometry* (Saxe et al., 2013) provides a more detailed characterization of the forward and backward signal propogation at initialization (Pennington et al., 2017; Hanin, 2018), enabling training 10,000-layer CNNs from scratch (Xiao et al., 2018). For residual networks, activation scale (Hanin & Rolnick, 2018), gradient variance (Balduzzi et al., 2017) and dynamical isometry property (Yang & Schoenholz, 2017) have been studied. Our analysis in Section 2 leads to the similar conclusion as previous work that the standard initialization for residual networks is problematic. However, our use of positive homogeneity for lower bounding the gradient norm of a neural network is novel, and applies to a broad class of neural network architectures (e.g., ResNet, DenseNet) and initialization methods (e.g., Xavier, LSUV) with simple assumptions and proof.

Hardt & Ma (2016) analyze the optimization landscape (loss surface) of linearized residual nets in the neighborhood around the zero initialization where all the critical points are proved to be global minima. Yang & Schoenholz (2017) study the effect of the initialization of residual nets to the test performance and pointed out Xavier or He initialization scheme is not optimal. In this paper, we give a concrete recipe for the initialization scheme with which we can train deep residual networks without batch normalization successfully.

**Understanding batch normalization.** Despite its popularity in practice, batch normalization has not been well understood. Ioffe & Szegedy (2015) attributed its success to "reducing internal covariate shift", whereas Santurkar et al. (2018) argued that its effect may be "smoothing loss surface". Our analysis in Section 2 corroborates the latter idea of Santurkar et al. (2018) by showing that standard initialization leads to very steep loss surface at initialization. Moreover, we empirically showed in Section 3 that steep loss surface may be alleviated for residual networks by using smaller initialization than the standard ones such as Xavier or He's initialization in residual branches. van Laarhoven (2017); Hoffer et al. (2018) studied the effect of (batch) normalization and weight decay on the effective learning rate. Their results inspire us to include a multiplier in each residual branch.

**ResNet initialization in practice.** Gehring et al. (2017); Balduzzi et al. (2017) proposed to address the initialization problem of residual nets by using the recurrence $\mathbf{x}_l = \sqrt{1/2}(\mathbf{x}_{l-1} + F_l(\mathbf{x}_{l-1}))$. Mishkin & Matas (2015) proposed a data-dependent initialization to mimic the effect of batch normalization in the first forward pass. While both methods limit the scale of activation and gradient, they would fail to train stably at the maximal learning rate for very deep residual networks, since

---

[6]For reference, we include a brief history of normalization methods in Appendix D.

they fail to consider the accumulation of highly correlated updates contributed by different residual branches to the network function (Appendix B.1). Srivastava et al. (2015); Hardt & Ma (2016); Goyal et al. (2017); Kingma & Dhariwal (2018) found that initializing the residual branches at (or close to) zero helped optimization. Our results support their observation in general, but Equation (5) suggests additional subtleties when choosing a good initialization scheme.

## 6 CONCLUSION

In this work, we study how to train a deep residual network reliably without normalization. Our theory in Section 2 suggests that the exploding gradient problem at initialization in a positively homogeneous network such as ResNet is directly linked to the blowup of logits. In Section 3 we develop Fixup initialization to ensure the whole network as well as each residual branch gets updates of proper scale, based on a top-down analysis. Extensive experiments on real world datasets demonstrate that Fixup matches normalization techniques in training deep residual networks, and achieves state-of-the-art test performance with proper regularization.

Our work opens up new possibilities for both theory and applications. Can we analyze the training dynamics of Fixup, which may potentially be simpler than analyzing models with batch normalization is? Could we apply or extend the initialization scheme to other applications of deep learning? It would also be very interesting to understand the regularization benefits of various normalization methods, and to develop better regularizers to further improve the test performance of Fixup.

### ACKNOWLEDGMENTS

The authors would like to thank Yuxin Wu, Kaiming He, Aleksander Madry and the anonymous reviewers for their helpful feedback.

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

## A  PROOFS FOR SECTION 2

### A.1  GRADIENT NORM LOWER BOUND FOR THE INPUT TO A NETWORK BLOCK

*Proof of Theorem 1.* We use $f_{i \to j}$ to denote the composition $f_j \circ f_{j-1} \circ \cdots \circ f_i$, so that $\mathbf{z} = f_{i \to L}(\mathbf{x}_{i-1})$ for all $i \in [L]$. Note that $\mathbf{z}$ is p.h. with respect to the input of each network block, i.e. $f_{i \to L}((1 + \epsilon)\mathbf{x}_{i-1}) = (1 + \epsilon)f_{i \to L}(\mathbf{x}_{i-1})$ for $\epsilon > -1$. This allows us to compute the gradient of the cross-entropy loss with respect to the scaling factor $\epsilon$ at $\epsilon = 0$ as

$$\frac{\partial}{\partial \epsilon} \ell(f_{i \to L}((1 + \epsilon)\mathbf{x}_{i-1}), \mathbf{y})\Big|_{\epsilon=0} = \frac{\partial \ell}{\partial \mathbf{z}} \frac{\partial f_{i \to L}}{\partial \epsilon} = -\mathbf{y}^T\mathbf{z} + \mathbf{p}^T\mathbf{z} = \ell(\mathbf{z}, \mathbf{y}) - H(\mathbf{p}) \qquad (6)$$

Since the gradient $L_2$ norm $\|\partial \ell / \partial \mathbf{x}_{i-1}\|$ must be greater than the directional derivative $\frac{\partial}{\partial t} \ell(f_{i \to L}(\mathbf{x}_{i-1} + t\frac{\mathbf{x}_{i-1}}{\|\mathbf{x}_{i-1}\|}), \mathbf{y})$, defining $\epsilon = t/\|\mathbf{x}_{i-1}\|$ we have

$$\left\|\frac{\partial \ell}{\partial \mathbf{x}_{i-1}}\right\| \geq \frac{\partial}{\partial \epsilon} \ell(f_{i \to L}(\mathbf{x}_{i-1} + \epsilon \mathbf{x}_{i-1}), \mathbf{y})\frac{\partial \epsilon}{\partial t} = \frac{\ell(\mathbf{z}, \mathbf{y}) - H(\mathbf{p})}{\|\mathbf{x}_{i-1}\|}. \qquad (7)$$

$\square$

## A.2 GRADIENT NORM LOWER BOUND FOR POSITIVELY HOMOGENEOUS SETS

*Proof of Theorem 2.* The proof idea is similar. Recall that if $\theta_{ph}$ is a p.h. set, then $\bar{f}^{(m)}(\theta_{ph}) \triangleq f(\mathbf{x}^{(m)}; \theta \setminus \theta_{ph}, \theta_{ph})$ is a p.h. function. We therefore have

$$\frac{\partial}{\partial \epsilon} \ell_{\text{avg}}(\mathcal{D}_M; (1+\epsilon)\theta_{ph}) \bigg|_{\epsilon=0} = \frac{1}{M} \sum_{m=1}^{M} \frac{\partial \ell}{\partial \mathbf{z}^{(m)}} \frac{\partial \bar{f}^{(m)}}{\partial \epsilon} = \frac{1}{M} \sum_{m=1}^{M} \ell(\mathbf{z}^{(m)}, \mathbf{y}^{(m)}) - H(\mathbf{p}^{(m)}) \quad (8)$$

hence we again invoke the directional derivative argument to show

$$\left\| \frac{\partial \ell_{\text{avg}}}{\partial \theta_{ph}} \right\| \geq \frac{1}{M \|\theta_{ph}\|} \sum_{m=1}^{M} \ell(\mathbf{z}^{(m)}, \mathbf{y}^{(m)}) - H(\mathbf{p}^{(m)}) \triangleq G(\theta_{ph}). \quad (9)$$

In order to estimate the scale of this lower bound, recall the FC layer weights are i.i.d. sampled from a symmetric, mean-zero distribution, therefore $\mathbf{z}$ has a symmetric probability density function with mean $\mathbf{0}$. We hence have

$$\mathbb{E}\ell(\mathbf{z}, \mathbf{y}) = \mathbb{E}[-\mathbf{y}^T(\mathbf{z} - \texttt{logsumexp}(\mathbf{z}))] \geq \mathbb{E}[\mathbf{y}^T(\max_{i \in [c]} z_i - \mathbf{z})] = \mathbb{E}[\max_{i \in [c]} z_i] \quad (10)$$

where the inequality uses the fact that $\texttt{logsumexp}(\mathbf{z}) \geq \max_{i \in [c]} z_i$; the last equality is due to $\mathbf{y}$ and $\mathbf{z}$ being independent at initialization and $\mathbb{E}\mathbf{z} = \mathbf{0}$. Using the trivial bound $\mathbb{E}H(\mathbf{p}) \leq \log(c)$, we get

$$\mathbb{E}G(\theta_{ph}) \geq \frac{\mathbb{E}[\max_{i \in [c]} z_i] - \log(c)}{\|\theta_{ph}\|} \quad (11)$$

which shows that the gradient norm of a p.h. set is of the order $\Omega(\mathbb{E}[\max_{i \in [c]} z_i])$ at initialization. $\square$

# B PROOFS FOR SECTION 3

## B.1 RESIDUAL BRANCHES UPDATE THE NETWORK IN SYNC

A common theme in previous analysis of residual networks is the scale of activation and gradient (Balduzzi et al., 2017; Yang & Schoenholz, 2017; Hanin & Rolnick, 2018). However, it is more important to consider the scale of actual change to the network function made by a (stochastic) gradient descent step. If the updates to different layers cancel out each other, the network would be stable as a whole despite drastic changes in different layers; if, on the other hand, the updates to different layers align with each other, the whole network may incur a drastic change in one step, even if each layer only changes a tiny amount. We now provide analysis showing that the latter scenario more accurately describes what happens in reality at initialization.

For our result in this section, we make the following assumptions:

- $f$ is a sequential composition of network blocks $\{f_i\}_{i=1}^{L}$, i.e. $f(\mathbf{x}_0) = f_L(f_{L-1}(\ldots f_1(\mathbf{x}_0)))$, consisting of fully-connected weight layers, ReLU activation functions and residual branches.
- $f_L$ is a fully-connected layer with weights i.i.d. sampled from a zero-mean distribution.
- There is no bias parameter in $f$.

For $l < L$, let $\mathbf{x}_{l-1}$ be the input to $f_l$ and $F_l(\mathbf{x}_{l-1})$ be a branch in $f_l$ with $m_l$ layers. Without loss of generality, we study the following specific form of network architecture:

$$
\begin{aligned}
F_l(\mathbf{x}_{l-1}) &= \overbrace{(\text{ReLU} \circ W_l^{(m_l)} \circ \cdots \circ \text{ReLU} \circ W_l^{(1)})}^{m_l \text{ ReLU}}(\mathbf{x}_{l-1}), \\
f_l(\mathbf{x}_{l-1}) &= \mathbf{x}_{l-1} + F_l(\mathbf{x}_{l-1}).
\end{aligned}
$$

For the last block we denote $m_L = 1$ and $f_L(\mathbf{x}_{L-1}) = F_L(\mathbf{x}_{L-1}) = W_L^{(1)}\mathbf{x}_{L-1}$.

Furthermore, we always choose 0 as the gradient of ReLU when its input is 0. As such, with input $\mathbf{x}$, the output and gradient of ReLU($\mathbf{x}$) can be simply written as $D_{\mathbb{1}[\mathbf{x}>0]}\mathbf{x}$, where $D_{\mathbb{1}[\mathbf{x}>0]}$ is a diagonal matrix with diagonal entries corresponding to $\mathbb{1}[\mathbf{x} > 0]$. Denote the preactivation of the $i$-th layer

(i.e. the input to the $i$-th ReLU) in the $l$-th block by $\mathbf{x}_l^{(i)}$. We define the following terms to simplify our presentation:

$$F_l^{(i-)} \triangleq D_{\mathbb{1}[\mathbf{x}_l^{(i-1)}>0]} W_l^{(i-1)} \cdots D_{\mathbb{1}[\mathbf{x}_l^{(1)}>0]} W_l^{(1)} \mathbf{x}_{l-1}, \quad l < L, i \in [m_l]$$

$$F_l^{(i+)} \triangleq D_{\mathbb{1}[\mathbf{x}_l^{(m_l)}>0]} W_l^{(m_l)} \cdots D_{\mathbb{1}[\mathbf{x}_l^{(i)}>0]}, \quad l < L, i \in [m_l]$$

$$F_L^{(1-)} \triangleq \mathbf{x}_{L-1}$$

$$F_L^{(1+)} \triangleq I$$

We have the following result on the gradient update to $f$:

**Theorem 3.** *With the above assumptions, suppose we update the network parameters by* $\Delta\theta = -\eta \frac{\partial}{\partial\theta}\ell(f(\mathbf{x}_0;\theta), \mathbf{y})$, *then the update to network output* $\Delta f(\mathbf{x}_0) \triangleq f(\mathbf{x}_0; \theta + \Delta\theta) - f(\mathbf{x}_0; \theta)$ *is*

$$\Delta f(\mathbf{x}_0) = -\eta \sum_{l=1}^{L} \left[ \sum_{i=1}^{m_l} \overbrace{\left\| F_l^{(i-)} \right\|^2 \left(\frac{\partial f}{\partial \mathbf{x}_l}\right)^T F_l^{(i+)} \left(F_l^{(i+)}\right)^T \left(\frac{\partial f}{\partial \mathbf{x}_l}\right)}^{\triangleq J_l^i} \right] \frac{\partial\ell}{\partial\mathbf{z}} + O(\eta^2), \quad (12)$$

*where* $\mathbf{z} \triangleq f(\mathbf{x}_0) \in \mathbb{R}^c$ *is the logits.*

Let us discuss the implecation of this result before delving into the proof. As each $J_l^i$ is a $c \times c$ real symmetric positive semi-definite matrix, the trace norm of each $J_l^i$ equals its trace. Similarly, the trace norm of $J \triangleq \sum_l \sum_i J_l^i$ equals the trace of the sum of all $J_l^i$ as well, which scales linearly with the number of residual branches $L$. Since the output $\mathbf{z}$ has no (or little) correlation with the target $\mathbf{y}$ at the start of training, $\frac{\partial\ell}{\partial\mathbf{z}}$ is a vector of some random direction. It then follows that the expected update scale is proportional to the trace norm of $J$, which is proportional to $L$ as well as the average trace of $J_l^i$. Simply put, to allow the whole network be updated by $\Theta(\eta)$ per step independent of depth, we need to ensure each residual branch contributes only a $\Theta(\eta/L)$ update on average.

*Proof.* The first insight to prove our result is to note that conditioning on a specific input $\mathbf{x}_0$, we can replace each ReLU activation layer by a diagonal matrix and does not change the forward and backward pass. (In fact, this is valid even after we apply a gradient descent update, as long as the learning rate $\eta > 0$ is sufficiently small so that all positive preactivation remains positive. This observation will be essential for our later analysis.) We thus have the gradient w.r.t. the $i$-th weight layer in the $l$-th block is

$$\frac{\partial\ell}{\partial\text{Vec}(W_l^{(i)})} = \frac{\partial\mathbf{x}_l}{\partial\text{Vec}(W_l^{(i)})} \cdot \frac{\partial f}{\partial\mathbf{x}_l} \cdot \frac{\partial\ell}{\partial\mathbf{z}} = \left(F_l^{(i-)} \otimes I_l^{(i)}\right) \left(F_l^{(i+)}\right)^T \frac{\partial f}{\partial\mathbf{x}_l} \cdot \frac{\partial\ell}{\partial\mathbf{z}}. \quad (13)$$

where $\otimes$ denotes the Kronecker product. The second insight is to note that with our assumptions, a network block and its gradient w.r.t. its input have the following relation:

$$f_l(\mathbf{x}_{l-1}) = \frac{\partial f_l}{\partial\mathbf{x}_{l-1}} \cdot \mathbf{x}_{l-1}. \quad (14)$$

We then plug in Equation (13) to the gradient update $\Delta\theta = -\eta\frac{\partial}{\partial\theta}\ell(f(\mathbf{x}_0;\theta), \mathbf{y})$, and recalculate the forward pass $f(\mathbf{x}_0; \theta + \Delta\theta)$. The theorem follows by applying Equation (14) and a first-order Taylor series expansion in a small neighborhood of $\eta = 0$ where $f(\mathbf{x}_0; \theta + \Delta\theta)$ is smooth w.r.t. $\eta$. $\qquad\square$

## B.2 WHAT SCALAR BRANCH HAS $\Theta(\eta/L)$ UPDATES?

For this section, we focus on the proper initialization of a scalar branch $F(x) = (\prod_{i=1}^{m} a_i)x$. We have the following result:

**Theorem 4.** *Assuming* $\forall i, a_i \geq 0, x = \Theta(1)$ *and* $\frac{\partial\ell}{\partial F(x)} = \Theta(1)$, *then* $\Delta F(x) \triangleq F(x; \theta - \eta\frac{\partial\ell}{\partial\theta}) - F(x; \theta)$ *is* $\Theta(\eta/L)$ *if and only if*

$$\left(\prod_{k\in[m]\setminus\{j\}} a_k\right) x = \Theta\left(\frac{1}{\sqrt{L}}\right), \quad \text{where} \quad j \in \arg\min_k a_k \quad (15)$$

*Proof.* We start by calculating the gradient of each parameter:

$$\frac{\partial \ell}{\partial a_i} = \frac{\partial \ell}{\partial F} \left( \prod_{k \in [m] \setminus \{i\}} a_k \right) x \tag{16}$$

and a first-order approximation of $\Delta F(x)$:

$$\Delta F(x) = -\eta \frac{\partial \ell}{\partial F(x)} (F(x))^2 \sum_{i=1}^{m} \frac{1}{a_i^2} \tag{17}$$

where we conveniently abuse some notations by defining

$$F(x) \frac{1}{a_i} \triangleq \left( \prod_{k \in [m] \setminus \{i\}} a_k \right) x, \quad \text{if } a_i = 0. \tag{18}$$

Denote $\sum_{i=1}^{m} \frac{1}{a_i^2}$ as $M$ and $\min_k a_k$ as $A$, we have

$$(F(x))^2 \cdot \frac{1}{A^2} \leq (F(x))^2 M \leq (F(x))^2 \cdot \frac{m}{A^2} \tag{19}$$

and therefore by rearranging Equation (17) and letting $\Delta F(x) = \Theta(\eta/L)$ we get

$$(F(x))^2 \cdot \frac{1}{A^2} = \Theta \left( \frac{\Delta F(x)}{\eta \frac{\partial \ell}{\partial F(x)}} \right) = \Theta \left( \frac{1}{L} \right) \tag{20}$$

i.e. $F(x)/A = \Theta(1/\sqrt{L})$. Hence the "only if" part is proved. For the "if" part, we apply Equation (19) to Equation (17) and observe that by Equation (15)

$$\Delta F(x) = \Theta \left( \eta(F(x))^2 \cdot \frac{1}{A^2} \right) = \Theta \left( \frac{\eta}{L} \right) \tag{21}$$

□

The result of this theorem provides useful guidance on how to rescale the standard initialization to achieve the desired update scale for the network function.

## C    ADDITIONAL EXPERIMENTS

### C.1    ABLATION STUDIES OF FIXUP

In this section we present the training curves of different architecture designs and initialization schemes. Specifically, we compare the training accuracy of batch normalization, Fixup, as well as a few ablated options: (1) removing the bias parameters in the network; (2) use 0.1x the suggested initialization scale and no bias parameters; (3) use 10x the suggested initialization scale and no bias parameters; and (4) remove all the residual branches. The results are shown in Figure 4. We see that initializing the residual branch layers at a smaller scale (or all zero) slows down learning, whereas training fails when initializing them at a larger scale; we also see the clear benefit of adding bias parameters in the network.

### C.2    CIFAR AND SVHN WITH BETTER REGULARIZATION

We perform additional experiments to validate our hypothesis that the gap in test error between Fixup and batch normalization is primarily due to overfitting. To combat overfitting, we use Mixup (Zhang et al., 2017) and Cutout (DeVries & Taylor, 2017) with default hyperparameters as additional regularization. On the CIFAR-10 dataset, we perform experiments with WideResNet-40-10 and on SVHN we use WideResNet-16-12 (Zagoruyko & Komodakis, 2016), all with the default hyperparameters. We observe in Table 4 that models trained with Fixup and strong regularization are competitive with state-of-the-art methods on CIFAR-10 and SVHN, as well as our baseline with batch normalization.

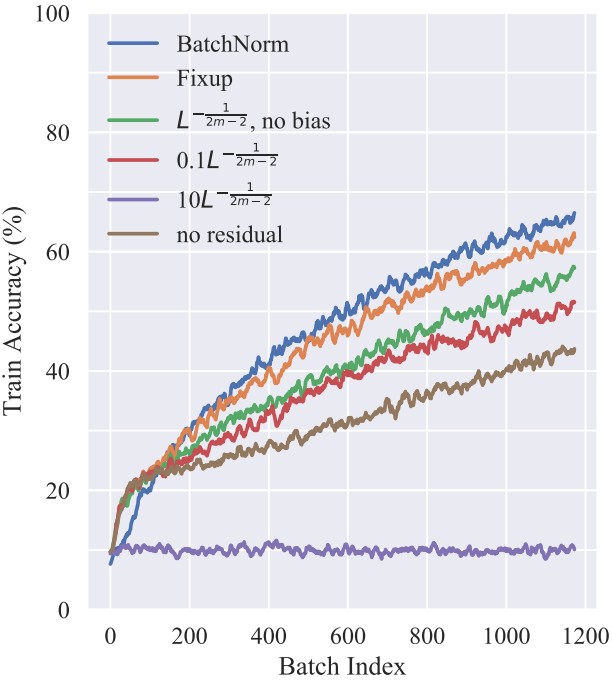

Figure 4: Minibatch training accuracy of ResNet-110 on CIFAR-10 dataset with different configurations in the first 3 epochs. We use minibatch size of 128 and smooth the curves using 10-step moving average.

| Dataset | Model | Normalization | Test Error (%) |
|---------|-------|---------------|----------------|
| CIFAR-10 | (Zagoruyko & Komodakis, 2016) | | 3.8 |
| | (Yamada et al., 2018) | Yes | **2.3** |
| | BatchNorm + Mixup + Cutout | | 2.5 |
| | (Graham, 2014) | No | 3.5 |
| | Fixup-init + Mixup + Cutout | | **2.3** |
| SVHN | (Zagoruyko & Komodakis, 2016) | | 1.5 |
| | (DeVries & Taylor, 2017) | Yes | **1.3** |
| | BatchNorm + Mixup + Cutout | | 1.4 |
| | (Lee et al., 2016) | No | 1.7 |
| | Fixup-init + Mixup + Cutout | | **1.4** |

Table 4: Additional results on CIFAR-10, SVHN datasets.

### C.3 TRAINING AND TEST CURVES ON IMAGENET

Figure 5 shows that without additional regularization Fixup fits the training set very well, but overfits significantly. We see in Figure 6 that Fixup is competitive with networks trained with normalization when the Mixup regularizer is used.

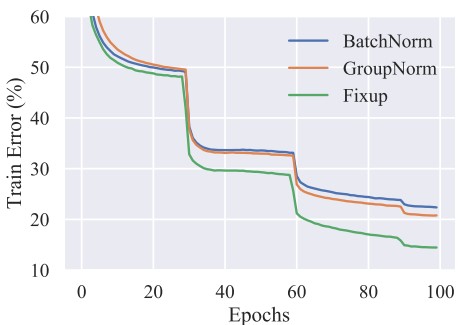
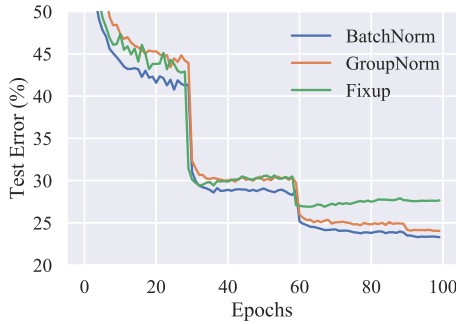

Figure 5: Training and test errors on ImageNet using ResNet-50 without additional regularization. We observe that Fixup is able to better fit the training data and that leads to overfitting - more regularization is needed. Results of BatchNorm and GroupNorm reproduced from (Wu & He, 2018).

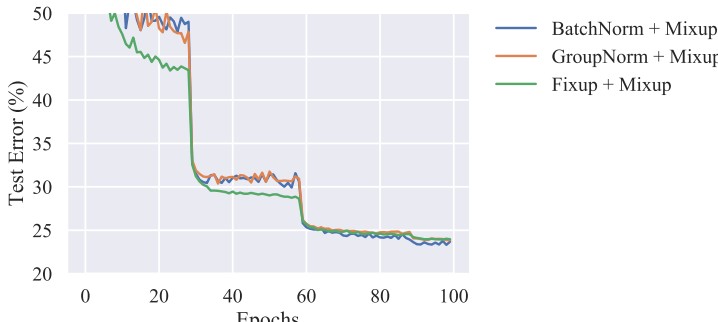

Figure 6: Test error of ResNet-50 on ImageNet with Mixup (Zhang et al., 2017). Fixup closely matches the final results yielded by the use of GroupNorm, without any normalization.

## D ADDITIONAL REFERENCES: A BRIEF HISTORY OF NORMALIZATION METHODS

The first use of normalization in neural networks appears in the modeling of biological visual system and dates back at least to Heeger (1992) in neuroscience and to Pinto et al. (2008); Lyu & Simoncelli (2008) in computer vision, where each neuron output is divided by the sum (or norm) of all of the outputs, a module called divisive normalization. Recent popular normalization methods, such as local response normalization (Krizhevsky et al., 2012), batch normalization (Ioffe & Szegedy, 2015) and layer normalization (Ba et al., 2016) mostly follow this tradition of dividing the neuron activations by their certain summary statistics, often also with the activation mean subtracted. An exception is weight normalization (Salimans & Kingma, 2016), which instead divides the weight parameters by their statistics, specifically the weight norm; weight normalization also adopts the idea of activation normalization for weight initialization. The recently proposed actnorm (Kingma & Dhariwal, 2018) removes the normalization of weight parameters, but still use activation normalization to initialize the affine transformation layers.

