# OpenReview forum: "Fixup Initialization: Residual Learning Without Normalization"
_ICLR.cc/2019/Conference_

### Official Review · AnonReviewer3 · 2018-11-01
**Interesting results. Normalization is not necessary to train deep resnets.**

**Rating:** 7
**Confidence:** 3

**Review:**


This paper shows that with a clever initialization method ResNets can be trained without using batch-norm (and other normalization techniques).  The network can still reach state-of-the-art performance.


The authors propose a new initialization method called "ZeroInit" and use it to train very deep ResNets (up to 10000 layers). They also show that the test performance of their method matches the performance of state-of-the-art results on many tasks with the help of strong data augmentation. This paper also indicates that the role of normalization in training deep resnets might not be as important as people thought. In sum, this is a very interesting paper that has novel contribution to the practical side of neural networks and new insights on the theoretical side.

Pros:
1. The analysis is not complicated and the algorithm for ZeroInit is not complicated.
2. Many people believe normalization (batch-norm, layer-norm, etc. ) not only improves the trainability of deep NNs but also improves their generalization. This paper provides empirical support that NNs can still generalize well without using normalization. It might be the case that the benefits from the data augmentation (i.e., Mixup + Cutout) strictly contain those from normalization. Thus it is interesting to see if the network can still generalize well (achieving >=95% test accuracy on Cifar10)  without using strong data-augmentation like mixup or cutout.
3.Theoretical analysis of BatchNorm (and other normalization methods) is quite challenging and often very technical. The empirical results of this paper indicate that such analysis, although very interesting, might not be necessary for the theoretical understanding of ResNets.


Cons:
1.The analysis works for positively homogeneous activation functions i.e. ReLU, but not for tanh or Swish.
2.The method works for Residual architectures, but may not be applied to Non-Residual networks (i.e. VGG, Inception)

---

> ### Author Response · Authors · 2018-11-13
> **Thanks; we totally agree**
>
> Dear AnonReviewer3, thank you for your encouraging review. We totally agree with your comments.
>
> A side note to your question: our experiments show that with standard data augmentation, the regularization effect of batch normalization can bring about 0.5% improvement in test accuracy on CIFAR-10, but we hypothesize some advanced regularization methods (such as ShakeDrop or DropBlock) could also make up for this gap.
>
> - References:
> [1] Yamada, Y., Iwamura, M., & Kise, K. (2018). ShakeDrop regularization. arXiv preprint arXiv:1802.02375.
> [2] Ghiasi, G., Lin, T. Y., & Le, Q. V. (2018). DropBlock: A regularization method for convolutional networks. arXiv preprint arXiv:1810.12890.

---

### Official Review · AnonReviewer1 · 2018-11-02
**The method presented is partially based on interesting observations, and it obtains good empirical results (tough not better than competition in general). However, the presentation is somewhat misleading: the method includes normalization elements not discussed, and some of its components are not justified and not tested empirically in isolation.**

**Rating:** 5
**Confidence:** 4

**Review:**

Summary:
A method is presented for initialization and normalization of deep residual networks. The method is based on interesting observations regarding forward and backward explosion in such networks with the standard Xavier or (He, 2015) initializations. Experiments with the new method show that it is able to learn with very deep networks, and that its performance is on a par with the best results obtained by other networks with more explicit normalization.
Advantages:
-	The paper includes interesting observations, resulting in two theorems,  which show the sensitivity of traditional initializations in residual networks
-	The method presented seems to work comparable to other state of the art initialization + normalization methods, providing overall strong empirical results.
Disadvantages:
-	The authors claim to suggest a method without normalization, but the claim is misleading: the network has additive and multiplicative normalization nodes, and their function and placement is at least as ‘mysterious’  as the role of normalization in methods like batch and layer normalization.
o	This significantly limits the novelty of the method: it is not ‘an intialization’ method, but a combination of initialization and normalization, which differ from previous ones in some details.
-	The method includes 3 components, of which only one is justified in a principled manner. The other components are not justified neither by an argument, nor by experiments. Without such experiments, it is not clear what actually works in this method, and what is not important.
-	The argument for the ‘justified’ component is not entirely clear to me. The main gist is fine, but important details are not explained so I could not get the entire argument step-by-step. This may be a clarity problem, or maybe indicate deeper problem of arbitrary decisions made without justification – I am not entirely sure. Such lack of clear argumentation occurs in several places
-	Experiments isolating the contribution of the method with respect to traditional initializations are missing (for example: experiments on Cifar10 and SVHN showing the result of traditional initializations with all the bells and whistles (cutout, mixup) as the zeroInit gets.

More detailed comments:
Page 3:
-	While I could follow the general argument before eq. 2, leading to the conclusion that the initial variance in a resnet explodes exponentially, I could not understand eq. 2. What is its justification and how is it related to the discussion before it? I think it requires some argumentation.
Page 4:
-	I did not understand example 2) for a p.h. set. I think an argument, reminder of the details of resnet, or a figure are required.
-	I could not follow the details of the argument leading to the zeroInit method:
o	How is the second design principle “Var[F_l(x_l)] = O( 1/L) justified?
As far as I can see, having Var[F_l(x_l)] = 1/L will lead to output variance of (1+1/L)^L =~e, which is indeed O(1). Is this the argument? Is yes, why wasn’t it stated? Also: why not smaller than O(1/L)?
o	Following this design principle several unclear sentences are stated:
	We strive to make Var[F_l(x_l)] = 1/L, yet we set the last convolutional layer in the branch to 0 weights. Does not it set Var[F_l(x_l)] = 0, in contradiction to the 1/L requirement?
	 “Assuming the error signal passing to the branch is O(1),” – what does the term “error signal” refer to? How is it defined? Do you refer to the branch’s input?
	I understand why the input to the m-th layer in the branch is O(\Lambda^m-1) if the branch input is O(1) but why is it claimed that “the overall scaling of the residual branch after update is O(\lambda^(2m-2))”? what is ‘the overall scaling after update’ (definition) and why is it the square of forward scaling?
-	The zero Init procedure step 3 is not justified by any argument in the proceeding discussion. Is there any reason for this policy? Or was it found by trial and error and is currently unjustified theoretically (justified empirically instead). This issue should be clearly elaborated in the text. Note that the addition of trainable additive and multiplicative elements is inserting the normalization back, while it was claimed to be eliminated. If I understand correctly, the ‘zeroInit’ method is hence not based on initialization (or at least: not only on initialization), but on another form of normalization, which is not more justified than its competitors (in fact it is even more mysterious: what should we need an additive bias before every element in the network?)
Page 5:
-	What is \sqrt(1/2) scaling? It should be defined or given a reference.
Page 6:
-	It is not stated on what data set figure 2 was generated.
-	In table 2, for Cifar-10 the comparison between Xavier init and zeroInit shows only a small advantage for the latter. For SVHN such an experiment is completely missing, and should be added.
o	It raises the suspect the the good results obtained with zeroInit in this table are only due to the CutOut and mixup used, that is: maybe such results could be obtained with CutOut+Mixup without zero init, using plain Xavier init? experiments clarifying this point are also missing.
Additional missing experiments:
-	It seems that  ZeroInit includes 3 ingredients (according to the box in page 4), among which only one (number 2) is roughly justified from the discussion.  Step 1) of zeroing the last layer in each branch is not justified –why are we zeroing the last layer and not the first, for example? Step 3 is not even discussed in the text – it appear without any argumentation. For such steps, empirical evidence should be brought, and experiments doing this are missing. Specifically experiments of interest are:
o	Using zero init without its step 3: does it work? The theory says it should.
o	Using only step 3 without steps 1,2. Maybe only the normalization is doing the magic?
The paper is longer than 8 pages.

I have read the rebuttal.
Regarding normalization: I think that there are at least two reasonable meanings to the word 'normalziation': in the wider sense is just means mechanism for reducing a global constant (additive normalization) and dividing by a global constant (multiplicative normalization). In this sense the constant parameters can be learnt in any way. In the narrow sense the constants have to be statistics of the data. I agree with the authors that their method is not normalization in sense 2, only in sense 1. Note that keeping the normalization in sense 1 is not trivial (why do we need these normalization operations? at least for the multiplicative ones, the network has the same expressive power without them).  I think the meaning of normalization  should be clearly explained in the claim for 'no  normalization'.
Regarding additional mathematical and empirical justifications required: I think such justifications are missing in the current paper version and are not minor or easy to add. I believe the work should be re-judged after re-submission of a version addressing the problems.

---

> ### Author Response · Authors · 2018-11-13
> **General reply (1): No normalization anywhere**
>
> Dear AnonReviewer1, we thank you for the very detailed review, and find it valuable for improving the writing of our paper in an updated version. We are happy to hear that you find our observations interesting, and our empirical results strong.
>
> Regarding your concerns:
> ------------------------------
>
>  -- The reviewer seems to think our method is "a combination of initialization and normalization".
>
> The proposed method does not use any normalization and so we believe there is a misunderstanding, either about the method, or about what is commonly regarded as normalization.
>
> We do not divide any neural network component by its statistics, neither do we subtract the mean from any activations. In fact, with our method there is **no computation of statistics (mean, variance or norm) at initialization or during any phase of training**.
>
> In a sharp contrast, all normalization methods for training neural networks explicitly normalize (i.e. standardize) some component (activations or weights) through dividing activations or weights by some real number computed from its statistics and/or subtracting some real number activation statistics (typically the mean) from the activations.
>
> To elaborate, we provide a brief historical background on normalization techniques. The first use of such ideas and terminology in modeling visual system dates back at least to Heeger (1992) in neuroscience and to Pinto et al. (2008) and Lyu & Simoncelli (2008) in computer vision, where each neuron output is divided by the sum (or norm) of all of the outputs, a module called divisive normalization. Recent popular normalization methods, such as local response normalization (Krizhevsky et al., 2012), batch normalization (Ioffe & Szegedy, 2015) and layer normalization (Ba et al., 2016) mostly follow this tradition of dividing the neuron activations by their certain summary statistics, often also with the activation mean subtracted. An exception is weight normalization (Salimans & Kingma, 2016), which instead divides the weight parameters by their statistics, specifically the weight norm; weight normalization also adopts the idea of activation normalization for weight initialization. The recently proposed actnorm (Kingma & Dhariwal, 2018) removes the normalization of weight parameters, but still use activation normalization to initialize the affine transformation layers.
>
> Therefore, our method is substantially different from all aforementioned techniques, and should not be regarded as being close to a normalization method.
>
> - References:
> [1] Heeger, D. J. (1992). Normalization of cell responses in cat striate cortex. Visual neuroscience, 9(2), 181-197.
> [2] Pinto, N., Cox, D. D., & DiCarlo, J. J. (2008). Why is real-world visual object recognition hard?. PLoS computational biology, 4(1), e27.
> [3] Lyu, S., & Simoncelli, E. P. (2008). Nonlinear image representation using divisive normalization. In IEEE Conference on Computer Vision and Pattern Recognition, 2008.
> [4] Krizhevsky, A., Sutskever, I., & Hinton, G. E. (2012). Imagenet classification with deep convolutional neural networks. In Advances in neural information processing systems (pp. 1097-1105).
> [5] Ioffe, S., & Szegedy, C. (2015). Batch normalization: Accelerating deep network training by reducing internal covariate shift. arXiv preprint arXiv:1502.03167.
> [6] Ba, J. L., Kiros, J. R., & Hinton, G. E. (2016). Layer normalization. arXiv preprint arXiv:1607.06450.
> [7] Salimans, T., & Kingma, D. P. (2016). Weight normalization: A simple reparameterization to accelerate training of deep neural networks. In Advances in Neural Information Processing Systems (pp. 901-909).
> [8] Kingma, D. P., & Dhariwal, P. (2018). Glow: Generative flow with invertible 1x1 convolutions. arXiv preprint arXiv:1807.03039.

---

> ### Author Response · Authors · 2018-11-13
> **General reply (2): further justifications**
>
>  -- The reviewer thinks that among the 3 components of ZeroInit, only Step 2 is justified in a principled manner. Step 1 and Step 3 are not justified by an argument or experiments.
>
> We will clarify the justification for each step in the paper. We hope you will find the following explanation helpful in understanding the effects and importance of each component. These improvements and new ablation experiments will appear in the revised paper.
>
> Summary: Step 2, combined with Step 1, ensures each SGD step updates the residual branch function by O(eta/L) so that the whole network is updated by O(eta). This is the most important component of our method and also distinguishes it from all previous work. Step 3 is indeed not essential for training, but the bias parameters (empirically) create better loss landscape, and the multipliers help us avoid tuning the global learning rate schedule.
>
> We now provide further in-depth justifications for each of the above arguments.
>
> Step 1 & 2:
>
> On one hand, as explained in the paper, initializing the residual branches to 0 prevents them from exploding and minimizes the lower bound of the gradient in Theorem 2. On the other hand, 0 initialization helps Step 2 limit the norm of the update of the residual branches to O(eta/L), as we now explain:
>
> Consider a residual branch with m layers, our goal is to derive the correct scaling for these layers, so that the residual branch is updated by O(eta/L) per gradient step. For simplicity, we assume the network is a composition of scalar functions (i.e. the input, output and hidden layers are all scalars), and there is no activation function. The residual branch can therefore be written as:
>
> F(x) = a_1 * ... * a_m * x
>
> where x is the input to this residual branch, and a_1, ..., a_m are nonnegative scalars (thinking of them as the rescaling of default initialization). Furthermore, we denote the gradient of the objective function w.r.t. F(x) as g. It is then easy to show that the gradient w.r.t. a_i is g * F(x) / a_i. Now if we perform a gradient descent update with step size eta, and calculate the update to F(x) using first-order approximation w.r.t. eta, we will get:
>
> \Delta F(x) =~ - eta * g * (F(x))^2 * ((1/a_1)^2 + ... + (1/a_m)^2)
>
> Note that we would like the scale of \Delta F(x) to be O(eta/L). Assuming g is O(1), it then follows that the scale of M = (1/a_1)^2 + ... + (1/a_m)^2 should be O(1/(L * (F(x))^2)). Let A = min_i {a_i} and we have (1/A)^2 <= M <= m * (1/A)^2. Put together, we arrive at A = O(sqrt{L} * F(x)). We hence finally get the desired design constraints:
>
> (I.) A = min_i {a_i},
> (II.) F(x) / A = O(1/sqrt{L})
>
> In sum, with (I.) and (II.) satisfied and assuming g is O(1), we can ensure the update of F(x) is O(eta/L), hence the update of the overall network is O(eta).
>
> A simple and natural design to satisfy these constraints is our Step 1 and Step 2. Furthermore, setting A to 0 (Step 1) has the additional benefit that each residual branch doesn't need to "unlearn" its random initial state, so that training proceeds faster in the first few epochs.
>
> Step 3:
>
> Using biases in the linear and convolution layers is a common practice in neural network history. In normalization methods, bias and scale parameters are typically used to restore the representation power after normalization. For example, in batch normalization gamma and beta parameters are used to affine-transform the normalized activations per each channel.
>
> Step 3 is the simplest design which provides similar representation power to affine layers. Our design is a substantial simplification of the common practice, in that we only introduce O(K) parameters beyond conv and linear weights (note that our conv and linear layers do not have biases), whereas the common practice includes O(KC) (e.g. batch normalization and weight normalization) or O(KCWH) (e.g. layer normalization) additional parameters, where K is the number of layers, C is the max number of channels per layer and W, H are the spatial dimension of the largest feature maps.
>
> Finally, it is important to note that the bias and multiplier parameters are not essential for training to proceed -- without them the training still works, even with 10,000 layers, albeit with suboptimal performance.

---

> > ### Public Comment · (anonymous) · 2018-11-15
> > **Re-ordering of multipliers and bias**
> >
> > Why were the biases and multipliers re-ordered, and one multiplier replaced with a bias (as in Figure 1)? The use of the architecture on the right of Figure 1 has still has not been justified over the (seemingly more natural) architecture in the middle of Figure 1.

---

> > > ### Author Response · Authors · 2018-11-15
> > > **It's actually (1) removing extra multiplier(s) and (2) adding biases before conv layers (i.e. after ReLU)**
> > >
> > > Thanks for asking!
> > >
> > > It may appear as we are doing a reordering, but in fact the right of Figure 1 makes two changes to the middle of Figure 1:
> > >
> > > (1) Deleting extra multiplier(s) so that there is only one multiplier per residual branch. This is because the effect of two (or more) multipliers is similar to that of one multiplier, which is to influence the effective learning rate of the conv layers in the same branch.
> > >
> > > (2) Adding a bias before each conv layer (i.e. changing ReLU-Conv to ReLU-Bias-Conv). The intuitive justification is that the preferred input mean of the conv layer may be different from the preferred output mean of the ReLU, hence a bias parameter allows for more representation power to satisfy both preferences. This is similar to why a bias term is added before ReLU (e.g. in standard feed-forward networks, Conv-BN-ReLU module, as well as our Conv-Bias-ReLU module).
> > >
> > > For additional justifications of (2), also note that there are debates about whether Conv-BN-ReLU or Conv-ReLU-BN is better in practice [1]; on the other hand, in [2, Figure 6 (d)] the authors find the best-performing residual branch to be "BN-Conv-BN-ReLU-Conv-BN". It may appear that the conclusion to draw from [2] is that one should use "more batchnorm and less relu [3]". However, if we remove the normalization layers in "BN-Conv-BN-ReLU-Conv-BN" and delete extra multipliers as per (1), we are left with:
> > > "Bias-Conv-Bias-ReLU-Conv-Multiplier-Bias",
> > > which is indeed very similar to what we proposed in the right of Figure 1:
> > > "Bias-Conv-Bias-ReLU-Bias-Conv-Multiplier-Bias".
> > >
> > > ------------------
> > > A side remark: when comparing middle and right of Figure 1, it may be helpful to switch the "bias" after the "+" into the residual branch, i.e. after the "multiplier", as the correspondence is easier to see this way and these two computation graphs are mathematically equivalent.
> > >
> > > ------------------
> > > References:
> > > [1] Batch Normalization before or after ReLU?https://www.reddit.com/r/MachineLearning/comments/67gonq/d_batch_normalization_before_or_after_relu/
> > > [2] Han, D., Kim, J., & Kim, J. (2017). Deep pyramidal residual networks. CVPR.
> > > [3] Andrej Karpathy. https://twitter.com/karpathy/status/827644920143818753?lang=en

---

> > > > ### Public Comment · (anonymous) · 2018-11-28
> > > > **Re:**
> > > >
> > > > Agreed, the correspondence is clearer when the bias is drawn in the residual branch instead of after the +. I saw that you just revised the manuscript, but you could consider making this change as well (since there is no real reason to draw it after the + instead of in the residual branch).
> > > >
> > > > Also, as a minor comment, the "√L" in the diagram ("scaled down by √L") is a different color and font than the "scaled down by."

---

> > > > > ### Author Response · Authors · 2018-11-28
> > > > > **Thanks for the suggestions**
> > > > >
> > > > > We agree. We will place this bias module inside the residual branch in the next revision.
> > > > >
> > > > > Also thank you for noting this detail -- should definitely be corrected :)

---

> ### Author Response · Authors · 2018-11-13
> **Reply to more detailed comments**
>
> Page 3:
> Eq. 2 is essentially restating the reasoning and conclusion before it in a mathematical way. It can be derived by calculating the variance of both the LHS and RHS of Eq. 1 and applying the independence assumption. The second equality can be shown by mathematical induction. We will clarify in the updated version.
>
> Page 4:
> - Thanks, we will add a figure to clarify each p.h. set example.
> - Yes, the fact "(1+1/L)^L =~e" is exactly why we would like the update of each residual branch rather than Var[F_l(x_l)] to be O(eta/L). Thanks for asking, we will correct in the updated version.
> - By "error signal" we mean the partial derivative of the loss function w.r.t. a layer. This term is used in e.g. (Schraudolph, 1998) but we now realize it is not clear. We will clarify its meaning.
> - Thanks for asking -- this is central to understanding our method. Please refer to our new analysis in justifying Step 1 & 2 above.
> - Please see the above general reply for justifications of step 3. Once again, we emphasize that our method is an initialization with minimal network components for achieving state-of-the-art performance, and contains no normalization operation.
>
> Page 5:
> - "\sqrt(1/2) scaling" is rescaling the activations by \sqrt(1/2) after each block. It is proposed as a possible remedy for ResNet without batch normalization in (Balduzzi et al., 2017).
>
> Page 6:
> - The dataset is CIFAR-10, as stated in the figure caption.
> - While the difference of the end performance of the two initialization is not huge (7% relative improvement for the median of 5 runs), we note that there is substantial difference in the difficulty of training. Network with ZeroInit is trained with the same learning rate and converge as fast as network trained with batch normalization, while we fail to train a Xavier initialized ResNet-110 with 0.1x maximal learning rate. Personal communication with the authors of (Shang et al., 2017) confirms our observation, and reveals that the Xavier initialized network need more epochs to converge.
> - Cutout and Mixup both contribute to the final performance in the CIFAR and SVHN experiments, as they likely supersede the regularization benefits of batch normalization. However, training with Xavier initialization cannot generalize as well, mainly because a substantially smaller learning rate has to be used to stabilize training, which in turn hurts generalization. We empirically validate this claim in the updated version.
> - We answered these questions in the general reply. In short, which layer to zero does not matter, training without step 3 works (though a bit worse). Using step 3 alone will not work due to incorrect scaling of the updates. We will add these experiments in the appendix.
>
> - References:
> [1] Schraudolph, N. N. (1998). Centering neural network gradient factors. In Neural Networks: Tricks of the Trade (pp. 207-226). Springer, Berlin, Heidelberg.
> [2] Balduzzi, D., Frean, M., Leary, L., Lewis, J. P., Ma, K. W. D., & McWilliams, B. (2017). The Shattered Gradients Problem: If resnets are the answer, then what is the question?. arXiv preprint arXiv:1702.08591.

---

### Official Review · AnonReviewer2 · 2018-11-02
**Interesting, but unsure about the impact**

**Rating:** 7
**Confidence:** 3

**Review:**

This paper proposes an exploration of the effect of normalization and initialization in residual networks. In particular, the Authors propose a novel way to initialize residual networks, which is motivated by the need to avoid exploding/vanishing gradients. The paper proposes some theoretical analysis of the benefits of the proposed initialization.

I find the paper well written and the idea well executed overall. The proposed analysis is clear and motivates well the proposed initialization. Overall, I think this adds something to the literature on residual networks, helping the reader to get a better understanding of the effect of normalization and initialization. I have to admit I am not an expert on residual networks, so it is possible that I have overlooked at previous contributions from the literature that illustrate some of these concepts already. Having said that, the proposal seems novel enough to me.

Overall, I think that the experiments have a satisfactory degree of depth. The only question mark is on the performance of the proposed method, which is comparable to batch normalization. If I understand correctly, this is something remarkable given that it is achieved without the common practice of introducing normalizations. However, I have not found a convincing argument against the use of batch normalization in favor of ZeroInit. I believe this is something to elaborate on in the revised version of this paper, as it could increase the impact of this work and attract a wider readership.

---

> ### Author Response · Authors · 2018-11-13
> **Comparison with previous work; practical implications**
>
> Dear AnonReviewer2, we appreciate your encouraging review and valuable suggestions. We hope to address your questions below:
>
> 1. The reviewer hopes to know if "previous contributions from the literature" have similar concepts.
>
> We listed related work we knew of by the time of paper submission. After submission, we did find more related work. Indeed, some previous works propose to initialize the residual branches in a way such that the network output variance is independent of depth, which is a necessary but not sufficient condition for training very deep residual networks, as we show in the updated version.
>
> However, none of the related work observes that the residual branches should be initialized in a way such that its update is O(eta/L) per SGD step, where eta is the maximal global learning rate and L is the total number of residual branches. This ensures the network has an update of O(eta) per SGD step, which we find is a sufficient condition for training to proceed as fast as batch normalization.
>
> 2. The reviewer has not found a "convincing argument against the use of batch normalization".
>
> Even if a practitioner continues to use batch normalization, we argue that this work helps understand how BatchNorm improves training.
>
> And for several tasks, batch normalization is not applicable or at least no preferable. Our method holds promise in many of these different tasks. For example, batch normalization is not used in many natural language tasks, where the state-of-the-art models use layer normalization (Vaswani et al., 2017), whereas we show our method can match or supercede its performance. In image super-resolution, it is recently shown that training without batch normalization improves performance (Lim et al., 2017); our method could possibly help achieve further improvement. In image style transfer, instance normalization is currently the standard technique (Ulyanov et al., 2016; Zhu et al., 2017); our method could possibly help as well. In semantic segmentation task, although batch normalization is found useful, its batchsize requirement put a severe constraint on the model size and the parallelizability of training, resulting in heavy burden of cross-GPU communication (Peng et al., 2017); hence using ZeroInit in combination with other regularization may be preferable. In image classification problems, current evidences are still in favor of batch normalization; however, as our method removes the necessity of using batch normalization in training and exposes the severe overfitting problem, future exploration of regularization methods that supersede batch normalization is possible.
>
> References:
> [1] Vaswani, A., Shazeer, N., Parmar, N., Uszkoreit, J., Jones, L., Gomez, A.N., Kaiser, Ł. and Polosukhin, I., (2017). Attention is all you need. In Advances in Neural Information Processing Systems (pp. 5998-6008).
> [2] Lim, B., Son, S., Kim, H., Nah, S., & Lee, K. M. (2017, July). Enhanced deep residual networks for single image super-resolution. In The IEEE conference on computer vision and pattern recognition (CVPR) workshops (Vol. 1, No. 2, p. 4).
> [3] Dmitry Ulyanov, Andrea Vedaldi, Victor Lempitsky. (2016). Instance Normalization: The Missing Ingredient for Fast Stylization
> [4] Zhu, J. Y., Park, T., Isola, P., & Efros, A. A. (2017). Unpaired image-to-image translation using cycle-consistent adversarial networks. arXiv preprint.
> [5] Peng, C., Xiao, T., Li, Z., Jiang, Y., Zhang, X., Jia, K., ... & Sun, J. (2017). Megdet: A large mini-batch object detector. arXiv preprint arXiv:1711.07240, 7.

---

> > ### Comment · AnonReviewer2 · 2018-11-22
> > **Thanks for the rebuttal**
> >
> > Many thanks for the rebuttal. After reading this and the other reviews, I'd be inclined to keep my score to "accept".

---

### Public Comment · (anonymous) · 2018-11-09
**Tensorflow implementation of resnet already has zero initialization by default; Comment on prior works**

Dear authors.

Thanks for an interesting paper. Incidentally, in the current resnet implementation (at least in TPU) in Tensorflow, the last batchnorm going back into the main branch as \gamma initialized to 0, which I believe achieves a similar effect to what you are doing here, at least from an initialization perspective. This has been around since February of this year.

https://github.com/tensorflow/tpu/blob/master/models/official/resnet/resnet_model.py#L219

Is the resnet in your experiments initialized like so? If not, how does such initialization compare to your initialization (without BN)?

In addition, please correct me if I'm mistaken, but the theoretical analysis of variances in this paper seems to have been done (quite thoroughly) in Yang & Schoenholz 2017 and Hanin & Rolnick 2018, where the theory in the former works for any nonlinearity and predicts the empirical results (for tanh and relu) highly accurately, while the latter mathematically characterizes the activation dynamics. The former paper is missing in the citation, while the latter only gets a passing mention. Could you comment on the novelty of the derivation in the current work and why it's not enough to use results from these two papers?

Yang & Schoenholz 2017 https://arxiv.org/abs/1712.08969
Hanin & Rolnick 2018 http://arxiv.org/abs/1803.01719

Thanks, and looking forward to your reply.

---

> ### Author Response · Authors · 2018-11-13
> **comparison with prior methods and theoretical work**
>
> Hi, thanks for your interest and pointer to related work! We believe that both our method and the theoretic analysis contain substantial novelty.
>
> A comparison with the gamma=0 alternative:
>
> For the batchnorm implementation, as the other comment pointed out, the suggestion of setting gamma=0 in the last batchnorm dates back at least to (Goyal et al., 2017). We agree that it is a great observation. However, setting gamma=0 for the last batchnorm is not sufficient for training without using a normalization method. As we explain in the paper, only setting the residuals to zero, the Step 1 of our method, will still result in explosion after a few steps. This is why our method requires Step 2 to lead to reliable convergence in all cases we tested.
>
> We summarize some key differences in the following, and also provide a detailed account of why the alternative method of setting gamma=0 would not work. For further information, please also refer to our reply to AnonReviewer1.
>
> The critical insight for our design is that, we would like to ensure the norm of the update to each residual branch function to be O(eta/L) per each step where eta is the maximal learning rate and L is the number of residual branches, hence ensuring the logits do not blow up after O(1/eta) steps. As we show in the updated version, a scalar ResNet model may help understand the argument.
>
> Step 2, combined with Step 1, ensures each SGD step updates the residual branch function by O(eta/L) so that the whole network is updated by O(eta). This is the most important component of our method and also distinguishes it from all previous work.
>
> For example, suppose the affine layers in batchnorm is preserved while the normalization layers are removed, and suppose we set gamma=0 in the last affine layer of each residual branch. What will happen in the first SGD update? By chain rule and Kaiming initialization, one can show that the gamma(s) in the last affine layer of each residual branch will get an update of O(eta), whereas the other layers in the residual branch get no updates. It then follows that each residual branch is a function of scale O(eta) after the first SGD update. Furthermore, we can show that all the residual branches are highly correlated after one update, resulting in output logits of O(1 + eta*L) scale, which leads to gradient explosion if L is large and eta is not small, as shown in our analysis.
>
> A comparison with related theoretic work:
>
> First, we thank you for bringing (Yang & Schoenholz 2017) to our attention. We appreciate the depth and mathematical skills demonstrated in both works, and agree that our analysis does not apply to arbitrary activation functions. That said, we would like to emphasize that our analysis excels in three aspects when compared with related work: general, realistic and simple. We now explain below:
>
> Generality:
>
> We only make two assumptions: (1) positive homogeneity and (2) weight distribution of the fully-connected layer. No other assumptions about the network structure is made (in particular, our analysis applies to (i) both the basic residual block and the bottleneck residual block; (ii) both the original version and the pre-activation version). No assumption about the distribution of other weights is made (in particular, our analysis applies to orthogonal initialization as well as data-dependent initialization).
>
> In contrast, Yang & Schoenholz (2017) only analyzed what they called the "reduced residual network" and the "full residual network", both of which only contains one activation function per each residual branch, hence does not apply to the usual 2-layer block or the bottleneck structure. Their analysis also requires both (Axiom 3.1) symmetry of activation and gradients and (Axiom 3.2) gradient independence. Finally, their analysis does not include convolutional layers, which are a crucial element of practical networks.
>
> Reality:
>
> With our general and mild assumptions, our analysis directly applies to the models and algorithms people implement.
>
> In contrast, the gradient independence assumption (Axiom 3.2) in (Yang & Schoenholz 2017) requires the forward and backward process to be fully decoupled, which is not the case for the networks that are used in practice.
>
> Simplicity:
>
> In addition to applying to real-world networks, our proof technique is simple and only involves basic probability and calculus. Our proof length is less than one page. In contrast, the proofs in (Yang & Schoenholz 2017) involve intricate algebraic manipulations and advanced math topics such as the mean field theory, and often span multiple pages.
>
> We would like to note that by all means we sincerely respect the works of (Yang & Schoenholz 2017) and (Hanin & Rolnick 2018), and will discuss their contributions in the revised paper. On the other hand, we also believe that simple and general theories such as our analysis are good things to have and to build upon.

---

> > ### Public Comment · (anonymous) · 2018-11-27
> > **Thanks for your response.**
> >
> > Hi, thanks for your response.
> >
> > Regarding gamma=0 BN networks, I agree there is some theoretical motivation for your method compared to the Goyal et al. method. However, I would still be very curious to see the result of comparing to gamma=0 BN networks empirically, i.e repeat your suite of tests with the standard resnet but just initialize BN gamma = 0. Also, if your analysis is correct, that there can be problems if eta and L are both large, then why can't one just scale eta as 1/L, at least initially in training?
> >
> > Regarding your comments on Yang & Schoenholz (2017): Correct me if I'm wrong, but the "Axiom 3.1" of that paper seems only assumed for nice presentation. "Axiom 3.2" (gradient independence) indeed seems unreasonable a priori, but as demonstrated in many papers by now (Schoenholz et al. 2017, Xiao et al. 2018, Karakida et al. 2018, Amari et al. 2018, and so on), this assumption leads to highly accurate predictions of gradient norms and other quantities. So while I agree you do not assume certain things in your paper, you also do not get prediction for the mean gradient norms and other quantities that can be verified. Thus claiming "generality" in this scenario seems misleading. In terms of measuring and correcting for gradient explosion, for example, I would think it's much better to get mean predictions of gradient norms rather than bounds which could be vacuous.
> >
> > Schoenholz, Gilmer, Ganguli, Sohl-Dickstein 2017. Deep Information Propagation
> > Xiao, Bahri, Sohl-Dickstein, Schoenholz, Pennington 2018. Dynamical Isometry and a Mean Field Theory of CNNs: How to Train 10,000-Layer Vanilla Convolutional Neural Networks
> > Karakida, Akaho, Amari 2018. Universal Statistics of Fisher Information in Deep Neural Networks: Mean Field Approach
> > Amari, Karakida, Oizumi 2018. Fisher Information and Natural Gradient Learning of Random Deep Networks

---

> > > ### Author Response · Authors · 2018-11-28
> > > **Could you please clarify the first part? Reply to the second part.**
> > >
> > > Thanks for your comments!
> > >
> > > One of the authors here. I think you raised interesting questions in the first part, but am not sure what you mean exactly there. Am I correct that you would like to:
> > > (1) see the result of a standard ResNet (i.e. with batch normalization layers) if we initialize the last gamma in each residual branch as 0;
> > > and (2) know if we can (or why we cannot) train a residual network with standard initialization and no normalization by setting eta as 1/L?
> > >
> > > Regarding the second part, indeed Yang & Schoenholz (2017) provide a more detailed characterization of the gradient norms and other quantities, which we very much appreciate. By "generality" we mean our analysis in Section 2 applies to different weight initialization schemes (e.g. not necessarily i.i.d.; can even be data-dependent) except for the i.i.d. assumption on the last fully-connected layer, whereas previous work typically assumes some particular initialization scheme (e.g. Yang & Schoenholz (2017) studied i.i.d. Gaussian weight initialization).
> > >
> > > On the other hand, our result in Section 2 does have limitations compared with Yang & Schoenholz (2017), in that it is a lower bound of gradient norm for certain layers. While it explains why gradient explosion happens in standard initialization, it does not tell us when gradient explosion is guaranteed to NOT happen, which is addressed in Yang & Schoenholz (2017) (though with additional assumptions).
> > >
> > > That said, the main message we hope to convey (in Section 3 and Appendix B) is that when studying multi-layer neural networks, it may be more important to think about the scale of function update than the scale of gradients (though of course they are related). Similar analysis for multi-layer linear networks is present in e.g. (Arora et al., 2018); and the study of maximal stable learning rate in (Saxe et al., 2013) may be another related finding. We believe this is a good way to study the optimization of deep neural networks.
> > >
> > > Arora, S., Cohen, N., & Hazan, E. (2018). On the optimization of deep networks: Implicit acceleration by overparameterization. arXiv preprint arXiv:1802.06509.
> > > Saxe, A. M., McClelland, J. L., & Ganguli, S. (2013). Exact solutions to the nonlinear dynamics of learning in deep linear neural networks. arXiv preprint arXiv:1312.6120.

---

### Public Comment · (anonymous) · 2018-11-10
**Prior work**

Hi,

This is an interesting paper. How would you compare your method to the method in [1] setting gamma=0 for every batchnorm going back to the main branch? On the surface the techniques look very similar and the authors in [1] also noted that such initialization improves optimization at the beginning of training.

[1] Goyal et al.  Accurate, Large Minibatch SGD: Training ImageNet in 1 Hour.

---

> ### Author Response · Authors · 2018-11-13
> **comparison with prior work**
>
> Hi, thanks for your interest and pointer to related work! Goyal et al. (2017) made a great observation, however setting gamma=0 for the last batchnorm is not sufficient for training without using a normalization method. As we explain in the paper, only setting the residuals to zero, the Step 1 of our method, will still result in explosion after a few steps. This is why our method requires Step 2 to lead to reliable convergence in all cases we tested.
>
> We summarize some key differences in the following, and also provide a detailed account of why the alternative method of setting gamma=0 would not work. For detailed justifications about Step 1 & 2, please refer to our "general reply (2)" to AnonReviewer1.
>
> The critical insight for our design is that, we would like to ensure the norm of the update to each residual branch function to be O(eta/L) per each step where eta is the maximal learning rate and L is the number of residual branches, hence ensuring the logits do not blow up after O(1/eta) steps. As we show in the updated version, a scalar ResNet model may help understand the argument.
>
> Step 2, combined with Step 1, ensures each SGD step updates the residual branch function by O(eta/L) so that the whole network is updated by O(eta). This is the most important component of our method and also distinguishes it from all previous work.
>
> Why simply setting gamma=0 does not work:
>
> Suppose the affine layers in batchnorm is preserved while the normalization layers are removed, and suppose we set gamma=0 in the last affine layer of each residual branch. What will happen in the first SGD update? By chain rule and Kaiming initialization, one can show that the gamma(s) in the last affine layer of each residual branch will get an update of O(eta), whereas the other layers in the residual branch get no updates. It then follows that each residual branch is a function of scale O(eta) after the first SGD update. Furthermore, we can show that all the residual branches are highly correlated after one update, resulting in output logits of O(1 + eta*L) scale, which leads to gradient explosion if L is large and eta is not small, as shown in our analysis.

---

> ### Public Comment · (anonymous) · 2019-02-06
> **Other prior work**
>
> This paper again shows the relevance of initialization and providing a proper variance flow through networks. This successfully allows to get rid of batch normalization without sacrificing performance. Although this work cites Glorot and He, it seems that they might have overseen that this idea originally stems from (LeCun et al., 1998). Also, this exact idea of replacing batch normalization with proper initialization and activation functions has already been presented for fully connected networks in (Klambauer et al., 2017). These seem to be two more relevant papers that have not been cited in this work.
>
> LeCun, Yann A., et al. "Efficient backprop." Neural networks: Tricks of the trade. Springer, Berlin, Heidelberg, 1998. 9-48.
> Klambauer, Günter, et al. "Self-normalizing neural networks." Advances in Neural Information Processing Systems. 2017.

---

### Public Comment · (anonymous) · 2018-11-20
**Code release**

Will you release the code for this paper? This would be helpful for reproducibility.

---

> ### Author Response · Authors · 2018-11-21
> **Yes, we will release the code**
>
> Thanks for asking! Yes, we will release the code after the review period.

---

> > ### Public Comment · (anonymous) · 2018-12-21
> > **code**
> >
> > just checking for an update on this i would love to use your method in my work!

---

> > > ### Author Response · Authors · 2019-03-16
> > > **An implementation of Fixup**
> > >
> > > Hi,
> > >
> > > Thanks for your interest in our work. A re-implementation based on our paper has been released at https://github.com/hongyi-zhang/Fixup
> > >
> > > Best,
> > > Hongyi, Yann and Tengyu

---

### Author Response · Authors · 2018-11-27
**Revised paper uploaded. New explanations and new results.**

Dear AC and anonymous reviewers,

Thanks for your helpful comments and suggestions! We have significantly revised the justification text of our method based on your feedback. While our method, experiments and existing analysis remain valid, we have added new results that we believe are worth noting:

(1) We provide a top-down analysis for motivating the proposed method (see Section 3). We make efforts to rewrite Section 3 and believe now we have convincing justifications to explain our empirical success.
(2) To support (1), we derive two new theorems (see Appendix B) which we believe shed new lights on the understanding of neural network training.
(3) We add an ablation study section (see Appendix C.1) to show each part of the proposed method play a role in the overall performance.
(4) We rewrite the related work section based on the feedback we get since the original submission. In particular, we (i) explain the difference between ZeroInit and normalization methods, (ii) compare our analysis in Section 2 with previous theoretical work, and (iii) compare our proposed method with previous ResNet initialization in practice.
(5) Empirical results on Transformer are slightly improved (see Table 3). We also include ResNet-101 results on ImageNet (see Table 2).

Thanks again for your attention! We are happy to take any questions.

---

### Meta-Review · Area_Chair1 · 2018-12-13

**Confidence:** 4
**Recommendation:** Accept (Poster)

**Metareview:**

The paper explores the effect of normalization and initialization in residual networks, motivated by the need to avoid exploding and vanishing activations and gradients. Based on some theoretical analysis of stepsizes in SGD, the authors propose a sensible but effective way of initializing a network that greatly increases training stability. In a nutshell, the method comes down to initializing the residual layers such that a single step of SGD results in a change in activations that is invariant to the depth of the network. The experiments in the paper provide supporting evidence for the benefits; the authors were able to train networks of up to 10,000 layers deep. The experiments have sufficient depth to support the claims. Overall, the method seems to be a simple but effective technique for learning very deep residual networks.

While some aspects of the network have been used in earlier work, such as initializing residual branches to output zeros, these earlier methods lacked the rescaling aspect, which seems crucial to the performance of this network.

The reviewers agree that the papers provides interesting ideas and significant theoretical and empirical contributions. The main concerns by the reviewers were addressed by the author responses. The AC finds that the remaining concerns raised by the reviewers are minor and insufficient for rejection of the paper.